# Injury-Related Behavioral Variables in Alpine Skiers, Snowboarders, and Ski Tourers—A Matched and Enlarged Re-Analysis

**DOI:** 10.3390/ijerph16203807

**Published:** 2019-10-10

**Authors:** Martin Niedermeier, Gerhard Ruedl, Martin Burtscher, Martin Kopp

**Affiliations:** Department of Sport Science, University of Innsbruck, Fürstenweg 185, 6020 Innsbruck, Austria

**Keywords:** sensation seeking, alpine skiing, ski touring, snowboarding

## Abstract

Behavioral variables might play an important role in explaining the differences in injury rates across winter sport disciplines and injury prevention programs might be more specifically designed based on this knowledge. On ski slopes, alpine skiing, snowboarding, and ski touring are the predominant winter sport disciplines. Therefore, the aim of the present study was to investigate possible differences in injury-related behavioral variables between practitioners of these disciplines. Using a matched re-analysis approach of a cross-sectional survey, 414 winter sport participants (alpine skiers, snowboarders, ski tourers, each *n* = 138) were analyzed on the differences in sensation seeking, treated injuries, and injury-related behavioral variables. Cochran–Mantel–Haenszel and Friedman tests revealed significantly higher sensation seeking, *p* < 0.001, and a significantly higher percentage of participants reporting to have consumed alcohol in the past five skiing days, *p* = 0.006, in snowboarders compared to alpine skiers. The participants with treated injuries showed higher sensation seeking, *p* < 0.050, and a higher percentage of snowboarders, *p* = 0.020, compared to participants without treated injuries. Injury prevention programs for snowboarders, who remain an important risk group for injury prevention, might benefit from considering a possibly higher percentage of alcohol-consuming participants and from providing information on injury-related risks of sensation seeking.

## 1. Introduction

Winter sports are increasingly popular and practiced by several hundred million people worldwide [1,2,3]. Three winter sport disciplines that are conducted on ski slopes are alpine skiing (AS), snowboarding (SB), and ski touring (ST). ST is traditionally done in free ski terrain, but the number of practitioners on ski slopes has increased in recent years [4]. ST uses a special boot fixing-system with artificial skins to walk up a mountain. Before skiing down on the slopes or in free ski terrain, the skin has to be removed and the boot fixing system has to be changed to a fixed mode. Out of the three disciplines, AS might be considered as the best studied discipline, where various potential health benefits were reported [5]. However, winter sports also bear an inherent risk for injuries. For AS, the injury risk is low when using a frequency (days of practice) to injury relation (approximately 1.3 injuries per 1000 skier days in Austria [6]). However, the absolute number of injuries per year remains high because of the large population at risk. The possibility of severe injury or death in the case of unexpected circumstances led to the classification high-risk sport for downhill winter sports in several studies, although this has been discussed controversially [7,8]. The injury rates in SB have decreased over time, however, they are still higher compared to skiers [9,10,11,12]. In slope ST, where usually a lower distance is skied compared to AS or SB [4], the data show an injury risk of 2.5 injuries per 1000 h of ST [13].

Risk-taking behavior is an important aspect of the research related to injury prevention [14]. For slope sports, several risk-taking behaviors, like speeding or participating under the influence of alcohol, have been specified [15,16]. On the other hand, it has to be stated that precautionary behavior, like wearing safety equipment, has considerably increased during the last decades [6,17]. The helmet usage rate in AS increased over the past years from 60% in 2010 [18], 65% in 2012 [19], 81/83% in 2015 [1], and 96% in 2019 [17]. Sensation seeking (SS) is one of the most studied underlying personality traits as it may be an important predictor when focusing on risk-taking behavior [20,21]. SS is linked to seeking out higher risk and it seems likely that aspects of SS would be related to risk-taking considering the excitement and risks of downhill sports [22]. SS is a “trait defined by the seeking of varied, novel, complex, and intense sensations and experiences, and the willingness to take physical, social, legal, and financial risk for the sake of such experience” [23]. SS is commonly assessed by a standardized self-report questionnaire allowing the interpretation of the four dimensions: thrill and adventure seeking, experience seeking, disinhibition, and boredom susceptibility [24,25].

In a previous study [26], the authors compared the above mentioned different winter sport disciplines on ski slopes (AS, SB, ST) with regard to SS based on the following arguments: Firstly, previous research recommended the SS scale as a useful tool to assess and interpret the individual differences in personality between people performing their sports with different levels of risk [22]. Secondly, SS seemed to be an important driver for risk behavior in skiers and SB [21,27,28,29]. Thirdly, different injury rates between different slope sports had not resulted in investigating these sports separately in personality-related research fields [27], even though there is evidence that motivational as well as personality-related aspects are involved when selecting a sport [20,30]. Lastly, the information on differences on injury-related behavioral variables between AS, SB, and ST might help to design injury prevention programs specifically for the disciplines.

Our previous analysis [26] has resulted in interesting differences in SS between AS, SB, and ST by a general comparison following an online survey using all datasets. In detail, the authors found higher SS in SB for the SS total score and all SS subscales compared to AS [26]. Similarly, higher scores were found in SB in the SS total score and in disinhibition compared to ST previously [26]. However, our previous analysis reported in [26] might have been limited by (non-significant) differences in the age class and sex of the practitioners. Comparing SB to ST, the percentage of males was 55% and 41%, respectively [26]. Comparing SB to AS, the percentage of 15–24 years was 56% and 50%, respectively [26]. However, both sex and age are discussed as strong factors for the differences in SS with men and adolescent people reporting the highest SS scores [31,32]. Consequently, even small, non-significant differences might have influenced our previous analysis. Furthermore, non-Bonferroni corrected post-hoc comparisons were used in the previous analysis, possibly resulting in a higher Type I error rate [33].

Therefore, the aims of the present study were to investigate possible differences in injury-related behavioral variables between alpine skiers, snowboarders and ski tourers and to critically re-assess whether the previously obtained results could withstand the use of a more rigid approach based on a matched design. Additionally, and as an enlarged analysis, the participants with and without injuries requiring medical care on ski slopes in the past were compared on possible differences in injury-related behavioral variables.

## 2. Materials and Methods

### 2.1. Procedure

The procedure and the questionnaire of the present cross-sectional study is described in detail in Kopp et al. [26]. Briefly, a web-based questionnaire (mean duration for completing: 15 min) was provided for the participants. The inclusion criteria were: (a) preferred winter sport either AS, or SB, or ST; and (b) age below 35 years. The Institutional Review Board of the Department of Sport Science at the University of Innsbruck provided ethical approval.

### 2.2. Questionnaire

The information on the following self-reported variables was collected: Sex (female or male), age class (15–24 or 25–34 years), preferred winter sport (AS, SB, or ST), winter sport ability (beginners, intermediates, advanced, or experts [34]), risk-taking behavior (more cautious or more risky [35]), helmet use (no or yes), alcohol consumption in the past five skiing days (never or at least once), treated injuries (never or at least once suffered from injuries requiring medical care on ski slopes), and SS (German version of the SS scale form V [36]). The SS scale consists of 40 items (response mode: two options, forced-choice) and allows for the calculation of a total score (sum of all items, lowest SS total score: 0, highest SS total score: 40) and four subscales (thrill and adventure seeking, disinhibition, experience seeking, and boredom susceptibility) ranging from 0 (lowest score) to 10 (highest score). The validity and reliability values of the SS scale were acceptable [22,37]. The Cronbach’s α values for the present sample (total SS: 0.76, thrill and adventure seeking: 0.67, disinhibition: 0.68, experience seeking: 0.51, boredom susceptibility: 0.40) were comparable to the values reported in a validity study of the German SS scale (total SS: 0.82, thrill and adventure seeking: 0.80, disinhibition: 0.69, experience seeking: 0.61, boredom susceptibility: 0.46) [36].

### 2.3. Statistical Analysis

SPSS Statistics version 25 (IBM, New York, NY, US) was used for statistical analyses. ST, matched AS, and matched SB were compared using a paired analysis approach [38]. Per each ST (*n* = 138), one matched AS (out of *n* = 726) and one matched SB (out of *n* = 321), were randomly selected using the software MedCalc (MedCalc Software, Ostend, Belgium). The matching criteria were sex (no deviation), age class (no deviation), and winter sport ability (no deviation). Exact matching for all three matching criteria was possible for all participants except *n* = 1 participant out of SB. Only sex and age were exactly matched for this participant, while winter sport ability differed. The matching criteria were selected based on significant correlation analyses between SS and associated factors in the total unmatched sample reported in Kopp et al. [26]. Cochran–Mantel–Haenszel tests were used to compare ST, AS, and SB on risk-taking behavior, helmet use, and alcohol consumption. As Shapiro-Wilk indicated the non-normal distribution of SS data, Friedman tests were used to analyze the differences in the SS total score for thrill and adventure seeking, disinhibition, experience seeking, and boredom susceptibility between AS, SB, and ST. In the case of significant findings, Bonferroni-corrected post-hoc tests (McNemar and Wilcoxon tests, as appropriate) were used for pairwise comparisons.

In addition, G*Power 3.1 (University of Düsseldorf, Dusseldorf, Germany) [39] was used to calculate the minimal effect sizes needed to be detected as significant in the pairwise comparisons. The idea behind this sensitivity analysis approach was to allow the comparison of significant results of both the present matched analysis and the previous unmatched analysis of the data, given the large discrepancy in the sample sizes (total *n* = 414 (in the matched analysis due to the paired analysis approach) versus total *n* = 1185 (in the unmatched analysis)). The settings in G*Power 3.1 for the present matched analysis were *p* = 0.017, power = 0.80, and *n* = 138. The settings for the unmatched analysis were *p* = 0.05, power = 0.80, *n*1 = 138/321, *n*2 = 321/726). A *p*-value of 0.050 was used because the Bonferroni correction was not applied in Kopp et al. [26]. The effect sizes were expressed as Cohens *d* [40].

For the comparison of participants with and without injuries requiring medical care on ski slopes in the past in the injury-related behavioral variables, Mann-Whitney U Tests were used for non-normally distributed SS variables and *χ*^2^ tests for categorical variables. *p*-values < 0.05 were considered to indicate statistical significance (two-tailed).

## 3. Results

In total, *n* = 414 participants were included in the matched analysis with *n* = 138 per each discipline (AS, SB, and ST). The sex distribution was 41% female and 59% male. The age distribution was 50% aged 15 to 24 years and 50% aged 25 to 34 years. Winter sport ability was 2% beginners, 12% intermediates, 65% advanced, and 20% experts (1% missing due to rounding). Winter sport ability for the SB participant, where exact winter sport ability matching was not possible, was advanced instead of the beginner.

### 3.1. Differences between Alpine Skiers, Snowboarders and Ski Tourers

Risk-taking behavior and helmet use were similar across AS, SB, and ST (Table 1). The percentages of those who reported alcohol consumption in the past five skiing days differed significantly across the disciplines. A post-hoc analysis showed that the percentage of those who reported no alcohol consumption in the past five skiing days was significantly lower in SB compared to AS, *p* = 0.014. The comparison between both SB versus ST and AS versus ST did not reach significance, *p* > 0.230.

There was a significant difference in the total score of SS across the disciplines (Table 2). SB reported a higher SS total score compared to AS, *p* = 0.006. The comparison between both SB versus ST and AS versus ST did not reach significance, *p* > 0.090. Out of the four subscales, the differences in disinhibition and experience seeking emerged as significant. The post-hoc analyses indicated a significant difference in disinhibition only, where SB reported higher values compared to both AS, *p* = 0.011, and ST, *p* < 0.001. The comparison between AS versus ST did not reach significance, *p* = 0.353. No significant differences between the disciplines were detected in the dimensions of thrill and adventure seeking as well as boredom susceptibility.

The sensitivity analyses for pairwise comparisons revealed a minimal effect size of Cohens *d* = 0.28 to be detected as significant in the matched post-hoc analysis. For the unmatched analysis, the sensitivity analyses for pairwise comparisons were calculated separately for each discipline. For SB versus ST/AS versus ST/SB versus AS, respectively, a minimal Cohens *d* = 0.29/0.27/0.19, respectively, was revealed to be detected as significant.

### 3.2. Comparison of Winter Sport Participants with and without Treated Injuries

Out of all participants, 96 participants (23%) reported to have suffered at least once from injuries requiring medical care on ski slopes in the past (Table 3). Significantly higher SS scores in participants with treated injuries were found for all SS variables, except thrill and adventure seeking compared to participants without treated injuries. A significantly higher percentage in participants with treated injuries reported to be more-risky compared to participants without treated injuries. The proportion of SB was significantly higher in participants with treated injuries compared to participants without treated injuries.

The predominant injury location was the lower leg for both ST and AS and the hand for SB (Table 4). The hip and upper leg was the least frequently mentioned injury location in all disciplines.

## 4. Discussion

The primary aim of the present study was to re-analyze possible differences in injury-related behavioral variables including SS between AS, SB, and ST in a matched analysis. The main results were that SB scored higher in the SS total score and the subscale disinhibition compared to ST and/or AS. Furthermore, the frequency of SB reporting to have consumed alcohol in the past five skiing days was significantly higher compared to AS. As a secondary aim, the participants with and without injuries requiring medical care on ski slopes in the past were compared. Significantly higher SS, a higher proportion of self-rated risk-taking behavior, as well as a higher percentage of SB were found in participants with treated injuries compared to participants without treated injuries.

### 4.1. Critical Comparison to Previous Analysis

In our previous analysis, a significantly higher mean value in SB was found for the SS total score compared to both AS and ST [26]. SB also reported higher scores in all SS subscales compared to AS previously. Compared to ST, higher SS in SB were found for thrill and adventure seeking, disinhibition, and experience seeking previously. These results could only partly be confirmed in the present matched analysis: The SS total score was significantly higher in SB compared to AS (but not to ST) and disinhibition was higher in SB compared to both AS and ST. The matching procedure resulted in unchanged mean SS scores of ST (since the smallest group was ST and therefore identical in both analysis), relatively unchanged mean SS scores of the SB in all domains, and higher mean SS scores in all domains for AS.

The higher mean scores of AS in all SS scores (for total SS: 21.8 vs. 21.1) were surprising since less males (41% instead of 55%) were included in the analysis. Given the well-known reported sex differences in SS, i.e., higher SS values in males [31], lower SS scores were expected in AS. The higher mean SS scores in AS are probably related to the matching criterion winter sport ability. The percentage of the advanced category in AS was 65% in the matched approach compared to 56% in the unmatched approach. Higher risk-taking was reported in higher skilled skiers [1]. Nevertheless, AS still showed a significantly lower total SS score and disinhibition compared to SB in the present matched analysis. These differences in SS might explain reported conflicts between these disciplines conducted on ski slopes to some extent [41]. SS in ST seems to be relatively similar to SS in AS. However, ST in the present study reflecting the slope tourers only, who might not be representative for ST in free ski terrain. In a previous study, higher SS scores were found in AS in free ski terrain (freeriders) compared to AS on ski slopes (slope skiers) [8]. Similar differences could be expected for ST in free ski terrain compared to ST on ski slopes.

Similar to the higher mean SS scores of AS, the relatively unchanged mean SS scores (for total SS: 23.7 vs. 23.7) in SB after the matching procedure were unexpected. A lower percentage of males (43%) and of people aged 15 to 24 years (50%) were included in present matched analysis compared to the previous unmatched analysis (55% males and 56% people aged 15 to 24 years). Both men and adolescent people showed the highest SS scores previously [28,29]. Consequently, lower SS scores were expected for the matched analysis. The SS scores of SB have to be considered high in comparison to a total mean score of 23.0 in high-risk sports participants (hang-gliders, mountaineers, sky-divers, automobile racers) and especially, a total mean score of 20.3 in low-risk sports participants (golfers, swimmers, marathon runners, aerobics) [42].

The finding that disinhibition remained the only subscale, where SB scored significantly higher compared to both other disciplines, was consistent in both analyses and seems to be a robust finding. Disinhibition is described as the seeking of sensation through drinking, partying, gambling and sexual variety [25]. The discipline-specific differences in disinhibition are backed up to some extent by significant differences in the percentages of alcohol-consuming participants while practicing the sport. SB reported more frequently to have consumed alcohol in the past five skiing days compared to AS, thereby mirroring the meaning of the subscale. Alcohol consumption was connected to winter sports-related injuries previously [43]. The higher SS in SB may reflect an underestimation of injury risks compared to skiers, bearing in mind that risk-taking behavior can be considered at least as associated with SS [21].

### 4.2. Comparison of Winter Sport Participants with and without Treated Injuries

The participants with injuries requiring medical care on ski slopes in the past showed significantly higher SS scores (with the exception of thrill and adventure seeking) and reported more risk-taking compared to participants without treated injuries. In other sports, it was proposed that people with specific personality traits (i.e., higher SS) seem to continue their risk-taking behavior and might be more prone to injuries [44,45]. More winter-sport-specific and in accordance, a higher rate of falls in ST with injury (35%) compared to ST without injury (15%) was reported previously [4]. It might be possible that people do not change their behavior following an injury (i.e., are more cautious after an injury). Instead, these people seem to continue their more-risky behavior. In contrast to Ruedl et al. [4], the presence of a previous injury was associated with *decreased* odds for injuries in park SB [46]. However, injuries in park SB are often associated with severe consequences [47], thereby possibly providing a more eminent trigger to change behavior compared to injuries with less severe consequences. In addition, the population differences of park SB [46,47] and recreational winter sport participants might account for the conflicting findings.

A higher percentage of SB was found in those with treated injuries (45%) compared to those without treated injuries (30%). In accordance, a higher prevalence of treated injuries in SB was reported in previous studies with what might be connected to a higher rate of falls in SB compared to skiers [9,10,11,12]. The discipline-specific injury locations of the present study are comparable to previous studies where 53% of all injuries were located in the lower limbs in AS compared to 46% located in the upper limbs in SB [12]. Similarly, 41% of all injuries were located in the knee in AS compared to 38% located in the upper limbs in SB [48].

The combination of these findings might be considered as highly practically relevant. Based on the present analyses and previous findings, SB remain an important risk group for injury prevention. Injury prevention programs [49] should specifically emphasize on providing information on injury-related risks fostered by the tendencies for disinhibited behavior patterns when focusing on SB. Generally, injury prevention programs might benefit from taking into account discipline-specific differences in motivational aspects associated with personality traits like SS [30,44,45].

### 4.3. Limitations

A few limitations have to be considered when interpreting the results of the present study. Firstly, the identical data set of our previous analysis [26] was used for the present study. This multiple inference approach might result in a higher Type I error rate [33] and needs replication in another data set. Secondly, the matching procedure introduced the possibility of overmatching [50], i.e., sex, age class, and winter sport ability were used as exact matching criteria and therefore, discipline-specific differences were not accounted for in the analysis. Connected with the matching procedure, it is questionable, if the discrepancies in the results of the matched and unmatched analysis are triggered by the matching procedure itself or by the differences in the sample size. The sensitivity analysis provided a differentiated picture. For the SB versus ST comparison, a smaller effect size was necessary to be detected as significant in the matched analysis compared to the unmatched analysis, indicating that the differences in the sample size plays a minor role in the discrepancies. However, for the AS versus ST/SB versus AS comparison, a larger effect size was necessary to be detected as significant in the matched analysis compared to the unmatched analysis. Therefore, the authors cannot exclude that the discrepancies in the results of the matched and unmatched analysis are driven by sample size differences, even though this seems unlikely given the higher mean SS values of AS in the matched (total SS: 21.8) compared to the unmatched analysis (total SS: 21.1). Thirdly, all limitations connected to a cross-sectional study based on self-reports have to acknowledged (e.g., impossible to assess causal relationships, non-truthfully answered questions, or a potential recall bias).

## 5. Conclusions

The present study indicated higher disinhibition scores in snowboarders compared to alpine skiers and ski tourers and a higher percentage of alcohol-consuming participants in snowboarders compared to alpine skiers, when sex, age, and winter sport ability were matched across the winter sport disciplines. Some previous findings [26] could be confirmed, thereby strengthening the advice to focus on disinhibited behavior patterns (e.g., alcohol consumption) in injury prevention programs for snowboarders. After controlling for age, sex, and winter sport ability through matching, sensation seeking in alpine skiers and ski tourers on ski slopes was widely similar. Therefore, discipline-specific differences in sensation seeking across winter sport participants are recommended to be considered between snowboarders and alpine skiers, but seem to play a minor role between alpine skiers and ski tourers. Furthermore, the higher disinhibition scores and self-reported higher percentage of alcohol-consuming participants in snowboarders might help to explain the higher injury prevalence compared to skiers [9,10,11,12]. Based on the comparison of winter sport participants with and without treated injuries, people with higher sensation seeking scores as well as people describing themselves as more risk-taking, and again, snowboarders should be addressed when developing injury prevention programs for winter sport disciplines conducted on ski slopes.

## Figures and Tables

**Table 1 ijerph-16-03807-t001:** Characteristics of the study participants separately for alpine skiers, snowboarders, and ski tourers.

Variable	Alpine Skiers		Snowboarders		Ski Tourers		*χ*^2^ (2) ^a^	*p*-Value ^a^
	(*n* = 138)		(*n* = 138)		(*n* = 138)			
	%	(*n*)	%	(*n*)	%	(*n*)		
Risk-taking behavior (missing cases *n* = 6)								
More cautious	56%	(74)	55%	(72)	62%	(82)		
More risky	44%	(58)	45%	(60)	38%	(50)	2.00	0.368
Helmet use (missing cases *n* = 1)								
No	33%	(45)	34%	(47)	40%	(55)		
Yes	67%	(92)	66%	(90)	60%	(82)	1.91	0.385
Alcohol consumption in the past five skiing days (missing cases *n* = 3)								
Never	73%	(99)	55%	(74)	63%	(85)		
At least once	27%	(36)	45%	(61)	37%	(50)	**10.35**	**0.006**

^a^ according to the Cochran–Mantel–Haenszel test. Bold values indicate significant differences.

**Table 2 ijerph-16-03807-t002:** Sensation seeking separately for ski tourers, alpine skiers, and snowboarders.

Variable	Alpine Skiers		Snowboarders		Ski Tourers		*χ*^2^ (2) ^a^	*p*-Value ^a^	Post-Hoc Analysis ^b^
	(*n* = 138)		(*n* = 138)		(*n* = 138)				
Sensation seeking total score									
Mean (SD)	21.8	(5.4)	23.7	(5.3)	22.3	(5.4)			
Median (IQR)	22.0	(18–26)	24.0	(20–28)	22.0	(18–27)	**10.45**	**0.005**	AS:SB
Thrill and adventure seeking									
Mean (SD)	6.9	(2.3)	7.2	(2.2)	7.4	(2.0)			
Median (IQR)	7.0	(6–9)	8.0	(6–9)	8.0	(6–9)	4.28	0.118	none
Disinhibition									
Mean (SD)	5.0	(2.3)	5.9	(2.2)	4.6	(2.4)			
Median (IQR)	5.0	(3–7)	6.0	(4–7)	5.0	(3–6)	**23.15**	**<0.001**	ST:SB, AS:SB
Experience seeking									
Mean (SD)	6.0	(1.7)	6.6	(1.8)	6.5	(1.7)			
Median (IQR)	6.0	(5–7)	6.0	(5–8)	7.0	(5–8)	**6.16**	**0.046**	none
Boredom susceptibility									
Mean (SD)	3.9	(1.8)	4.0	(1.7)	3.8	(2.0)			
Median (IQR)	4.0	(3–5)	4.0	(3–5)	4.0	(3–5)	0.60	0.742	none

AS: alpine skiers, SB: snowboarders, ST: ski tourers, SD: standard deviation, IQR: interquartile range, ^a^ according to the Friedman test, ^b^ according to Bonferroni-corrected pairwise comparisons (significant differences specified, e.g., AS:SB). Bold values indicate significant differences.

**Table 3 ijerph-16-03807-t003:** Injury-related behavioral variables separately for participants with and without injuries requiring medical care on ski slopes.

Variable	Without Treated Injuries		With Treated Injuries		z ^a^	*p*-Value ^a^
	(*n* = 317)		(*n* = 96)			
	Mean	(SD)	Mean	(SD)		
Sensation seeking (missing cases *n* = 1 each)						
Sensation seeking total score	22.1	(5.3)	24.2	(5.4)	**−3.14**	**0.002**
Thrill and adventure seeking	7.1	(2.2)	7.6	(2.0)	−1.91	0.056
Disinhibition	5.3	(2.2)	6.0	(1.9)	**−2.65**	**0.008**
Experience seeking	5.9	(2.1)	6.4	(2.2)	**−1.96**	**0.050**
Boredom susceptibility	3.8	(1.8)	4.2	(1.9)	**−1.97**	**0.048**
	**%**	**(*n*)**	**%**	**(*n*)**	***χ*^2^^b^**	***p*-value ^b^**
Risk-taking behavior (missing cases *n* = 7)						
More cautious	61%	(191)	46%	(44)		
More risky	39%	(121)	54%	(51)	**6.63**	**0.010**
Helmet use (missing cases *n* = 2)						
No	38%	(120)	29%	(28)		
Yes	62%	(196)	71%	(68)	2.48	0.115
Alcohol consumption in the past five skiing days (missing cases *n* = 4)						
Never	65%	(204)	60%	(58)		
At least once	35%	(110)	40%	(38)	0.66	0.416
Winter sport discipline (missing cases *n* = 1)						
Alpine skiers	34%	(108)	30%	(29)		
Snowboarders	30%	(95)	45%	(43)		
Ski tourers	36%	(114)	25%	(24)	**7.83**	**0.020**

SD: standard deviation, ^a^ according to the Mann-Whitney U Test, ^b^ according to the *χ*^2^ test. Bold values indicate significant differences.

**Table 4 ijerph-16-03807-t004:** Frequencies of injured anatomical locations separately for alpine skiers, snowboarders, and ski tourers (participants with treated injuries only).

Injured Anatomical Location	Alpine Skiers		Snowboarders		Ski Tourers	
	(*n* = 29)		(*n* = 43)		(*n* = 24)	
	%	(*n*)	%	(*n*)	%	(*n*)
Head/face	28%	(8)	19%	(8)	21%	(5)
Shoulder/upper arm	21%	(6)	23%	(10)	13%	(3)
Elbow/forearm	14%	(4)	16%	(7)	13%	(3)
Hand	28%	(8)	49%	(21)	29%	(7)
Trunk/spine	7%	(2)	26%	(11)	13%	(3)
Hip/upper leg	7%	(2)	2%	(1)	4%	(1)
Lower leg/ankle/foot	48%	(14)	33%	(14)	50%	(12)

Multiple responses allowed.

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
