# Peer review of "Injury-Related Behavioral Variables in Alpine Skiers, Snowboarders, and Ski Tourers—A Matched and Enlarged Re-Analysis"

_ijerph, 2019, doi:10.3390/ijerph16203807_

Round 1
Reviewer 1 Report
I thank the Editor and authors for the opportunity to review this interesting manuscript.
An understanding of risk taking propensity is useful when designing intervention to reduce injury in adventure sports. This study reports a re-analysis of a sub-set of previously-reported cross-sectional data. My comments now follow.
With respect for the likelihood that English is not the author’s primary language, I do suggest the authors re-consider use of the word “accident”, which is now banned/dissuaded by a number of public health journals. It is used in the abstract (pg 1, lines 16, 19 and 20), and main text (pg 3, line 40, pg 5 lines 7,9, 10, 12, and elsewhere).
Pg 3, lines 44+, since there are a little over 100 people in each adventure sport, I think reporting percentages (e.g. 41.3% female) to one decimal place might imply an accuracy greater than might be found in the data.
Regards the matching, the authors selected a ratio of 1:1:1 but then included one case where matching was not possible for all three matching criteria. Personally, I would have chosen to exclude this person. There were probably more than one match from AS and SB to the n=138 ST. Can the authors please add a brief note to state how they selected the matched data when there were duplicate matches? E.g., if there were three SB with the same age, sex and experience as a particular ST, how did they select which of those three would be included?
Table 1 – My personal preference is to report the n and have the % in parentheses. The authors may wish to check the preferred format of this journal.
Table 4 – Without meaning to belabour the point made earlier, I do not know what the authors mean by “accidents”. I left some equipment at home once by accident, also I’ve taken the wrong turn by accident and gotten lost, also I’ve been injured. If the participants were asked if they had suffered any injuries then these findings would be of more public health interest than if they were asked if they had “had any accidents”. On page 6, line 9, it suggests the inclusion criteria were “an accident requiring medical care” which I’d suggest could be simplified to “treated injuries”.
The Discussion is well thought-out and was interesting to read.
The limitations section on page 8 neglects to mention other potential limitations of cross sectional research, for example that associations may highlight further topics for research but do not imply causality. This has a bearing on such statements as page 7 lines 14/15, commenting on “..a more cautious behaviour following accidents.”. The self-reported “accidents” may have not been caused by errant behaviour.
Also, the discussion in the limitations section around the differences in results between the unmatched and matched analyses need not, in my opinion, be considered a limitation. The strength of matching is in the ability to control for known risk factors, such as age and sex. Therefore, it might be described as less of a limitation and more of a strength that this matched study has accounted for these known risk factors and identified that there are still significant differences between types of winter sports participants.
Page 8, Conclusions: In short, SB drink alcohol more frequently during ski-days than the other two groups, they have higher SS total, and higher disinhibition. There was no difference between AS and ST in SS total, or disinhibition (ST were slope tourers only though), and after matching on age, sex and skill all three groups had similar scores for thrill and adventure seeking, or boredom susceptibility. Line 13, instead of “Contrary to previous findings…” I would suggest perhaps consider “After controlling for age, sex and ability through matching…”.
References: Appear appropriate, mostly modern and in reputable journals.
Once again, I thank the Editor and authors for the opportunity to review this interesting manuscript.
Reviewer 2 Report
Page 1
Lines 33-34. This statement doesn’t make sense as it is incomplete, “discipline, where various potential health benefits [5],” Suggest finishing the thought and restructuring the entire sentence.
Lines 34-37. Rewrite the sentence as it is run-on and confusing. “Despite the injury risk of alpine skiing is low when using a frequency (days of practice) to injury relation (approximately 1.3 injuries per 1,000 skier days in Austria [6]), the absolute number of injuries per year remains high because of the large population at risk.”
Page 2
Lines 6-8. Sentence is confusing and needs to be rewritten. Example: Sensation seeking (SS) is one of the most studied underlying personality traits as it may be an important predictor when focusing on risk-taking behavior [20,21].
Lines 15-16. Rewrite sentence as its flow is confusing.
Line 23. Instead of “Fourthly” change to “Lastly”
Lines 25-26. Why are the acronyms not used in the text (AS, etc.)?
Lines 28-33. These sentences need to be merged and rewritten with the preceding sentences to make sense.
Page 3
Line 7. Remove sensation seeking and only need to use the acronym SS and add ( before German
Lines 12-15. These statements seem like they belong in the statistical analysis or results section based on how it currently reads. “The Cronbach’s α values for the present sample (total SS: .76, thrill and adventure seeking: .67, disinhibition: .68, experience seeking: .51, boredom susceptibility: .40) were comparable to values reported in a validity study of the German SS scale (total SS: .82, thrill and adventure seeking: .80, disinhibition: .69, experience seeking: .61, boredom susceptibility: .46) [36].”
Lines 21- 22. Change person to participant for word consistency. And apply this throughout the manuscript
Page 4
Line 47. I read the sentence as participants were classified into two age groups; 15-24 years and 25 to 34 years. The distribution is over a 20 year span but the authors make is sound as classified and was this analyzed?
Line 47. Sport ability is classified though there is no explanation in methods of how this decision is made.
Page 5
The tables need formatting for alignment (Specifically table 5), this may be corrected in the publication process.
Page 6
Lines 3-12. Use previously established acronyms
Overall Discussion section comment.
“Significantly higher sensation seeking, a higher proportion of self-rated risk-taking behavior, as well as a higher percentage of snowboarders were found in participants with accidents compared to participants without accidents.” The order in the discussion ends the first section with the previous sentence then shifts to previous data. An improved flow would occur with the Comparison of W/ & W/out accidents as it transitions more smoothly from the last sentence.
Line 33. The word should be “extent” not “extend.
Page 7.
Line 5. The word should be “extent” not “extend.
Lines 25-26. The total percentage is only 75% so where is the remaining 25% for snowboarders? “A higher percentage of snowboarders was found in those with accidents (45%) compared to those without accidents (30%).”
Reviewer 3 Report
It is well organized in a concise manner.
To prevent accidents, it is important to analyze environmental factors, improve skills, and research human factors. It is difficult to reduce sports accidents and research methods are also difficult. As described below, the authors have performed an appropriate analysis on the survey and obtained reasonable results.
Using a matched re-analysis approach of a cross-sectional survey, authors analyzed winter sport participants on differences in sensation seeking, accidents, and injury-related behavioral variables.
Authors show participants with accidents showed higher sensation seeking, and a higher percentage of snowboarders, compared to participants without accidents. Injury prevention programs for snowboarders, who remain an important risk group for injury prevention, might benefit from considering a possibly higher percentage of alcohol-consuming participants and from providing information on injury-related risks of sensation seeking.
